# An Exploration of the Coherent Effects between METTL3 and NDUFA10 on Alzheimer’s Disease

**DOI:** 10.3390/ijms241210111

**Published:** 2023-06-14

**Authors:** Lin Yang, Xinping Pang, Wenbo Guo, Chengjiang Zhu, Lei Yu, Xianghu Song, Kui Wang, Chaoyang Pang

**Affiliations:** 1College of Computer Science, Sichuan Normal University, Chengdu 610101, China; 2West China School of Basic Medical Sciences & Forensic Medicine, Sichuan University, Chengdu 610041, China

**Keywords:** Alzheimer’s disease, m6A, METTL3, NDUFA10

## Abstract

Alzheimer’s disease (AD) is a neurodegenerative disorder characterized primarily by a decline in cognitive function. However, the etiopathogenesis of AD is unclear. N6-methyladenosine (m6A) is abundant in the brain, and it is interesting to explore the relationship between m6A and AD causes. In this paper, the gene expression of METTL3 and NDUFA10 were found to correlate with the Mini-mental State Examination (MMSE), which is a clinical indicator of the degree of dementia. METTL3 is involved in post-transcriptional methylation and the formation of m6A. NDUFA10 encodes the protein with NADH dehydrogenase activity and oxidoreductase activity in the mitochondrial electron transport chain. The following three characteristics were observed in this paper: 1. The lower the expression level of NDUFA10, the smaller the MMSE, and the higher the degree of dementia. 2. If the expression level of METTL3 dropped below its threshold, the patient would have a risk of AD with a probability close to 100%, suggesting a basic necessity for m6A to protect mRNA. 3. The lower the expression levels of both METTL3 and NDUFA10, the more likely the patient would suffer from AD, implying the coherence between METTL3 and NDUFA10. Regarding the above discovery, the following hypothesis is presented: METTL3 expression level is downregulated, then the m6A modification level of NDUFA10 mRNA is also decreased, thereby reducing the expression level of NDUFA10-encoded protein. Furthermore, the abnormal expression of NDUFA10 contributes to the assembly disorder of mitochondrial complex I and affects the process of the electron respiratory chain, with the consequent development of AD. In addition, to confirm the above conclusions, the AI Ant Colony Algorithm was improved to be more suitable for discovering the characteristics of AD data, and the SVM diagnostic model was applied to mine the coherent effects on AD between METTL3 and NDUFA10. In conclusion, our findings suggest that dysregulated m6A leads to altered expression of its target genes, thereby affecting AD’s development.

## 1. Introduction

As one of the most common forms of dementia in elderly people, Alzheimer’s disease (AD) is a neurodegenerative disease [1]. Clinical manifestations of AD include symptoms such as cognitive impairment and memory loss [2,3,4]. The prevalence of AD is very high worldwide owing to a shortage of valid treatments [5].To date, the specific pathogenesis of AD remains unknown, probably because there is no single pathogenesis of AD.

Lately, it has been quite clear that RNA modifications act in regulating gene expression, which leads to dysfunction in many fundamental cellular processes in the pathogenesis of AD [6,7]. A highly abundant modification of RNA in the brain is m6A, with dynamic and reversible regulation by RNA methyltransferases, demethylases, and m6A binding proteins [8,9]. Numerous studies have demonstrated that m6A modifications can impact RNA fate and correlate with neurodevelopmental and neurodegenerative diseases [10,11,12]. In addition, mounting evidence suggests that m6A methylation modifications are involved in various biological processes, including stress response regulation [13], synaptic function [14], and cognitive function [11,15,16,17]. In neuroscience, m6A modifications may present a new frontier that could offer a novel view for our understanding of neurological disease mechanisms. However, the role of m6A regulators in AD and its related genes has been largely unexplored.

METTL3 is involved in post-transcriptional methylation and the formation of m6A. The function of METTL3 has been researched previously in the context of AD [18]. NDUFA10 encodes the protein with NADH dehydrogenase activity and oxidoreductase activity in the mitochondrial electron transport chain. The abnormal expression of NDUFA10 diminishes the efficiency of mitochondrial ATP synthesis [19,20]. It manifests as a disorder with tissue-specific neurological deficits that depend on the degree of mitochondrial dependence of brain tissue on ATP production [21,22].

We aimed to identify the most likely gene targeted by the methyltransferase METTL3 during m6A writing in AD. In this study, we first identified the m6A regulator METTL3, the reduced expression level of which is closely correlated with the evolution of AD. Furthermore, we comprehensively assessed the novel role of METTL3 in energy metabolism by modular exploration. NDUFA10 was screened from the modules by the AI Ant Colony Algorithm. NDUFA10 was not only the top-ranked among the most related gene set with METTL3 but also showed the best association with AD by fitting with MMSE. Eventually, we constructed a diagnostic model of AD via the combination of METTL3 and NDUFA10 and validated its classification effect. Our findings propose that NDUFA10 may be a potential key target for m6A methylation modification, and its reduction may be associated with AD through the m6A mechanism. More importantly, we suggest that m6A modification can regulate energy metabolism by affecting the electron respiratory chain and promote the onset and development of AD. Therefore, our study provides a novel prospect for diagnosing AD accurately.

## 2. Results

### 2.1. m6A Writer Gene METTL3 Is Downregulated in AD

To measure the aberrations of m6A in AD, twenty common m6A regulators were incorporated into the analysis from the published literature. Their expression levels were extracted from GSE5281. Differential expression analysis was performed between AD patients and controls. Among the 20 m6A regulators, m6A writer, eraser, and reader genes showed uneven expression (Figure 1A). METTL3, an m6A writer, was more lowly expressed in AD patients than in normal brains (Figure 1B). The expression level of METTL3 was also decreased in AD patients in comparison with normal individuals in GSE1297 (Figure 1C). In addition, accounting for the heterogeneity of brain tissue, the altered expression of METTL3 in GSE5281 was explored in different brain regions. The results revealed that METTL3 was notably downregulated in all brain regions involved compared to the controls (Figure 1D). These results tentatively suggested that METTL3 expression dropped abnormally in AD samples. This phenomenon may be associated with the aberrant modification of m6A, which in turn affects the onset and progression of AD.

### 2.2. The Set of Candidate AD Genes Positively Correlated with METTL3

To further identify METTl3-related AD genes, differentially expressed genes were screened from two aspects. On the one hand, 5858 genes differentially expressed (All_DEGs) between AD and normal samples were obtained in GSE5281. As shown in Figure 2A, there were 2685 up-regulated genes and 3173 downregulated genes in AD patients. On the other hand, on the basis of the median expression level of METTL3, AD patients were classified into two groups. Differential analysis was performed between METTL3_high and METTL3_low subgroups. In total, 5190 DEG genes (METTL3_DEGs) associated with METTL3 expression status were obtained (Figure 2B). The 2342 crossover genes were obtained by taking the overlap of the two DEGs sets (Figure 2C). We defined the intersection as METTL3-related AD genes, that is, a set of genes associated with METTL3 and affecting AD occurrence.

To find the key module most associated with METTL3 in AD, weighted gene correlation analysis (WGCNA) was performed on the above-mentioned METTL3-related AD genes, yielding the identification of nine modules (Figure 3A). To identify the modules relevant to METTL3 expression status, the relationship between each module and METTL3 expression status was analyzed (Figure 3B). Since METTL3 catalyzes the occurrence of m6A methylation modification to enhance transcript stability, we focused more on the module that was positively correlated with METTL3 expression, the blue module (r = 0.57, *p* < 0.001). The blue modular genes we found were the set of AD genes positively correlated with METTL3.

### 2.3. Functional Enrichment Analysis for the Genes That Positively Related to METTL3

In total, 417 genes positively associated with METTL3 expression were obtained from the blue module above. These genes were gathered together to form a set noted as $S_{Mettl3}$. In addition, the protein–protein interaction (PPI) network showed that $S_{Mettl3}$ could interact directly or indirectly with some AD genes (Figure 4A). To gain insight into the biological roles of $S_{Mettl3}$, GO and KEGG analyses were performed. GO enrichment analysis showed that $S_{Mettl3}$ were markedly enriched in biological processes such as ATP metabolic process and mitochondrial inner membrane (Figure 4B). Additionally, KEGG analysis suggested that $S_{Mettl3}$ were the most abundant in neurodegenerative multiple diseases and amyotrophic lateral sclerosis (Figure 4C).

### 2.4. The Effect of METTL3 on Energy Metabolism

To further discover the effect of METTL3 expression status on energy metabolism-related pathways, the ATP metabolic process, oxidative phosphorylation, mitochondrial ATP synthesis coupled electron transport, oxidoreductase complex, and NADH dehydrogenase complex pathways obtained from the above enrichment analysis were selected to be analyzed further. The genes in these five pathways were collected together and formed a set denoted as $S_{Energy-Mettl3}$. The set $S_{Energy-Mettl3}$ represented AD genes that were highly positively correlated with METTL3 expression and involved in the cellular process of energy metabolism. First, the expression of $S_{Energy-Mettl3}$ was visualized separately from the heat map. As shown in Figure 5A–E, the expression of any of the five pathway genes was decreased compared to the controls.

Then, the gene sequences were calculated in AD using the Ant Colony Algorithm on the basis of $S_{Energy-Mettl3}$ as well as m6A regulators. The calculation was iterated ten times to avoid coincidence. The results indicated that the genes appearing the highest number of times near METTL3 were SDHA and NDUFA10 (Table 1). This meant that an alteration of METTL3 would be accompanied by a shift in SDHA and NDUFA10. Finally, to verify the association between the selected genes and AD, the relationship between gene expression and clinical indicators was fitted by linear regression. Mini-mental State Examination (MMSE) and Neurofibrillary Tangle (NFT) are clinically implicated in the diagnosis of cognitive decline and dementia. Compared to SDHA, NDUFA10 was of more potential research significance (Figure 6A–D), namely as the candidate gene.

### 2.5. Co-Effects of METTL3 and NDUFA10 on AD

METTL3 represented a central member of the m6A modification. It counted in the transfer of methyl from the SAM complex to the target transcript. Following analysis by WGCNA, the Ant Colony Algorithm, and linear regression, the gene of the closest association with METTL3, NDUFA10, was identified in AD. Here, it was hypothesized that NDUFA10 may bind specifically to METTL3 as a target transcript in the form of m6A modification. To explore this mechanism, SRAMP was used to search for sites containing m6A modification in NDUFA10. As shown in Figure 7A,B, these sites were of high confidence. Such binding sites were valuable for subsequent experimental validation, possibly explaining the molecular mechanism of NDUFA10 in the etiopathogenesis of AD.

Then, to verify whether METTL3 and NDUFA10 were involved with AD, the expression data of both were used as input to build the SVM diagnostic model. As shown in Figure 8, totally two characteristics were observed: 1. There was a threshold in the development of AD for METTL3 expression levels. When the expression level was less than the threshold, the samples belonged to AD patients with a high probability close to 100%. 2. AD could be distinguished from normal brain samples by the combination of METTL3 and NDUFA10. Furthermore, when both METTL3 and NDUFA10 were highly expressed, where $X_{METTL3}$ was to the right of the black dashed line and $Y_{NDUFA10}$ was on the upper side of the yellow dashed line, individuals would be unlikely to develop into AD patients. In addition, the predictive performance of the model reached 0.915 as assessed by 10-fold cross-validation (Figure 9A–J).

## 3. Discussion

As a result of abnormal m6A modification, crucial cellular processes may be dysregulated, disrupting homeostasis and eventually causing related diseases [23]. Previous studies have revealed that m6A methylation modification is involved with AD [24]. Nearly half of neuronal mRNAs are m6A modified, and a decrease in m6A impairs neurogenesis and neuronal function [25]. However, few studies have examined the relationship between m6A regulators and AD. Several studies have demonstrated that the expression of m6A regulators varies between tissues and correlates with neurodegenerative processes [26]. These regulators can function in a spatiotemporal manner in early and late brain development and activate AD-related genes by controlling their protein levels [27]. METTL3, as an m6A writer, is a methyltransferase that regulates m6A modification in mRNA. However, in accordance with current studies, there is an under-explored mechanism of METTL3 in AD. This study attempts to integrate bioinformatics and statistical approaches to investigate METTL3-targeted transcripts and their biological processes in AD.

We identified the overall expression pattern of m6A regulators by differential expression analysis between AD brain and normal tissue. In comparison with controls, m6A writer genes were expressed at lower levels, while m6A eraser genes were expressed at higher levels in AD patients. m6A reader genes show uneven expression levels, suggesting that the stability of specific transcripts is usually restricted in AD brains [28]. Different patterns of these m6A regulators can lead to an aberrant post-transcriptional modification environment [29], which may alter the expression of AD-related genes. Specifically, we observed a significant downward trend in METTL3 in AD brain tissue. A decrease in METTL3 expression could mediate m6A dysregulation, which may contribute to neural degeneration in AD [18]. This finding may be a promising target for the exploration of AD causes. In addition, the accumulation of METTL3 was observed in the insoluble fraction, which is positively correlated with the level of insoluble Tau protein in postmortem human AD samples [30]. Tau enriches in neuronal axons, and the progress of Tau-targeted therapy in AD was described by Yi Guo’s study [31].

We further screened 2342 intersecting genes by differential expression analysis. These genes were implied to be the AD genes most associated with METTL3 differential expression. A METTL3-related module, the blue module (Figure 3B), was then identified from the intersecting genes according to the WGCNA method. The genes in this module showed the most relevance to the high-expressed state of METTL3. METTL3 is fundamentally associated with long-term memory formation in the human and mouse hippocampus [27]. The existence of different molecular networks between the module genes was confirmed by the PPI network. It is possible that the expression of certain AD genes may be altered by METTL3, thus contributing to the development of AD.

A total of 417 genes were obtained from the blue module. Their biological functions were explored by functional enrichment analysis. GO analysis showed that the module genes were enriched in ATP metabolic processes and mitochondrial inner membrane, etc. We focused the enrichment results on the pathways involved in the electron transport chain. For the five pathways screened, heat maps revealed that the expressions of these genes were all decreased in AD in comparison with controls. KEGG analysis showed that the aberrant expression of these genes correlated with the pathogenesis of various neurodegenerative diseases, including AD. This confirms the link between neurological disorders and energy metabolism in terms of data analysis. Furthermore, the module genes were screened based on their association with METTL3, implying a possible molecular network of METTL3-mediated methylation modifications and energy metabolism with neurological diseases. Animal experiments have shown that METTL3 was an essential regulator of postnatal development and energy homeostasis in iBAT. BAT-specific deletion of METTL3 resulted in significant downregulation of genes associated with developmental maturation, respiratory electron transport chain, adaptive thermogenesis, and energy deprivation [32].

Then, the gene sequence calculated by the AI Ant Colony Algorithm showed that SDHA and NDUFA10 had the highest association with METTL3. The gene sequence being formed by a global optimal alignment implied that METTL3 was globally aligned with SDHA and NDUFA10 at the time of AD progression [33,34]. Next, the relationship between these two genes and the clinical indicators MMSE and NFT was analyzed to investigate the association with AD. We found that when the expression level of NDUFA10 was lower, the MMSE was smaller. The results showed that NDUFA10 tended to be associated not only with METTL3 but also that its reduction would contribute to the onset and development of AD. Both expression levels were downregulated in AD brain tissue, implying that the patients were more susceptible to AD. NDUFA10 is an accessory subunit of the mitochondrial respiratory chain complex I. Mutations in subunits of complex I can cause a variety of defects, such as reactive oxygen species production, neurodegenerative diseases, apoptosis, and cell death [35,36,37]. Complex I assembly was disrupted due to mutations in NDUFA10 in individuals with Leigh syndrome [38]. Furthermore, the study by Ahmed-Noor A. Agip et al. [39] revealed the location of NDUFA10 in complex I and a nucleotide bound in subunit NDUFA10. In the electron transport chain of mitochondria, complex I is the first rate-limiting enzyme required for ATP production [40]. Mitochondrial complex I abnormalities are closely related to Tau load, reflecting the neuronal damage that occurs in the mild AD brain [41,42,43]. We suspect that in AD brain tissue, low expression of METTL3 leads to aberrant m6A modification, which is an upstream event for the downregulation of NDUFA10. More importantly, NDUFA10 affects the homeostatic level of complex I, resulting in issues in the electron respiratory chain. Several studies have linked reduced mitochondrial complex activity and abundance (especially complex I/IV) and bioenergetic imbalance (ROS/ATP) to AD [44,45,46,47,48,49]. Therefore, m6A may potentially exert an effect on the occurrence of AD by modifying NDUFA10. However, the exact mode of impact and its abnormal interactions in AD development need to be further explored.

Finally, we constructed a diagnostic model of AD using METTL3 and NDUFA10 to validate our speculation. The SVM diagnostic model reconfirmed that the combination of METTL3 and NUDFA10 had a significant role in the progression of AD. More importantly, the threshold of METTL3 expression level presents a promising novel thought for the future detection of AD. The cross-validation results showed a high AUC value. This implies that our model may provide a fresh perspective for the diagnosis of AD. In addition, the m6A modification mechanism between METTL3 and NDUFA10 was further validated by the SRAMP online site.

Taken together, m6A reader METTL3 holds a critical role in the evolution and progression of AD. Our study demonstrates that NDUFA10 may be its potential key methylation target. Low expression of METTL3 deprives part of the mRNA of base A of methylation protection, where A is the base adenine. Therefore, the amount of NDUFA10-encoded normal protein becomes too low, which reduces the efficiency of NADH dehydrogenase activity and oxidoreductase activity in the electron respiratory chain, thus affecting AD disease. These findings provide new insights for both prevention and intervention in AD.

## 4. Materials and Methods

### 4.1. Data Collection and Preprocessing

The Gene Expression Omnibus (https://www.ncbi.nlm.nih.gov/geo/, accessed on 8 March 2022) was used to download GSE5281 and GSE1297 gene expression data. GSE5281 obtained data from brain tissue sections using the GPL570 platform, including brain tissue from (1) the entorhinal cortex (2) the hippocampus (3) the medial temporal gyrus (4) the posterior cingulate gyrus (5) the superior frontal gyrus, and (6) the primary visual cortex. The dataset included 74 controls and 87 AD patients. The validation dataset GSE1297 was obtained using the GPL96 platform and included 22 AD patients (7 mild, 8 moderate, and 7 severe patients according to their degree of disease) and 9 normal hippocampal tissue samples. The “normalize Between Arrays” function in the “limma” package (version 3.48.3) was used to normalize gene expression profiles. In addition, MMSE and NFT data from AD patients were collected from GSE1297 and employed for further analysis.

### 4.2. Screening of METTl3-Related AD Genes

Twenty common m6A regulators were selected from the published literature as shown in Table 2. The expression patterns of these regulators during the developmental stages of AD were identified by analyzing the differences between AD patients and controls in GSE5281. On the other hand, significant alterations in METTL3 as an m6A writer gene in AD were identified through analysis of the differences between AD patients and controls in GSE5281 and GSE1297.

To identify the genes that vary with an alteration in METTL3, for the selected dataset (GSE5281), AD samples were divided into METTL3_high and METTL3_low subpopulations by selecting the median expression of METTL3 as the cutoff value. Differential expressed genes (DEGs) analysis was performed between AD patients and controls and between METTL3_high and METTL3_low subgroups by lmFit and eBayes methods. The “limma” package (version 3.48.3) was applied to analyze the differential expression. *p* < 0.05 was assumed to be significant. The intersection of the DEGs from both analyses was defined as the METTl3-related AD genes.

### 4.3. Identification of Key Module Based on the METTl3-Related AD Genes

The “WGCNA” package (version 1.71) was applied to identify the expression data of METTL3-related AD genes in GSE5281. Initially, the “hclust” function was applied to perform hierarchical clustering of AD samples and to determine the presence of outliers. Then, through the “pickSoftThreshold” function, the soft threshold power was calculated. Next, the “blockwiseModules” function was applied to construct co-expression network modules. Finally, in order to associate modules with features, a module mainly associated with METTL3_High or METTL3_Low subgroups was defined as the key module and selected for further screening.

### 4.4. PPI Network Construction

The “STRINGdb” package (version 2.4.2) was used to perform protein–protein interaction analysis of genes in the key module. First, interactions were screened by protein–protein interaction scores. Then, protein interaction information was obtained by the “get_interactions” function for subsequent visualization and analysis. Next, the “igraph” function was used to create network data and determine the network center by setting parameter information (number of nodes > 5). Finally, the “ggraph” function was used to plot the network data.

### 4.5. Functional Enrichment Analysis of Key Module Genes and Key AD Genes

GO and KEGG pathway analyses were performed through the “clusterProfiler” package enrichment function in R software (version 4.1.1) for genes in the key module through WGCNA analysis. *p* < 0.05 was considered as the cutoff criterion to target METTL3-related genes to certain biological functions. For GO enrichment results, genes in the most valuable pathways were selected as key AD genes.

### 4.6. Identification of Candidate Gene

The Ant Colony Algorithm was used to compute a gene sequence where genes with similar expression levels were aligned together. First, the distance between every two genes was calculated using the Euclidean square degree. Then, they were arranged according to their relevance to each other. The more relevant they are, the closer they are to each other. Then, the result of iteration to the maximum number was used as the gene sequence. Next, 10 gene sequences were obtained by repeating the calculation while counting the number of occurrences of each gene near METTL3. Those with more than 5 occurrences were considered to be the genes with the most similar expression level to METTL3. All calculations were dependent on Python (version 2.1).

A linear fit of the selected genes (in the most valuable pathways) to the MMSE, NFT data in GSE1297 was performed by the “lm” regression function. The best fit was selected as the gene associated with AD. Finally, the candidate gene was the best result of the above computational screenings.

### 4.7. The m6A Mechanism between METTL3 and Candidate Gene

The SRAMP online database (http://www.cuilab.cn/sramp, accessed on 6 September 2022) was applied to further verify whether the candidate gene and METTL3 can constitute an m6A-dependent mechanism. By predicting the precise location of m6A modifications on the candidate gene, SRAMP can provide a confidence level for each modification.

### 4.8. Construction and Validation of the Diagnostic Model for AD

A support vector machine (SVM) was applied in Python (version 2.1) using the “sklearn” package to build a diagnostic model. The model was able to distinguish AD from normal samples by the combination of METTL3 and the candidate gene. Samples from the GES5281 dataset were randomly assigned to the training set (60%) and the test set (40%). The area under the ROC curve (AUC) was applied to measure the merit of the diagnostic model. Ten AUC values were calculated by 10-fold cross-validation. Their mean value was taken as the predictive performance of the evaluation model. The higher the mean value, the more reliable the model.

## 5. Conclusions

In conclusion, METTL3 performs a vitally significant role in the onset and development of AD. More importantly, METTL3 is closely linked to the complex I subunit NDUFA10, which is a potential target gene for m6A methylation modification. This implies that m6A modification can regulate energy metabolism by affecting the electron respiratory chain in AD. Our findings provide novel insights into the pathogenesis of AD in the context of m6A methylation modifications. In future studies, more target genes of m6A methylation and its more detailed mechanisms in AD occurrence need to be further explored.

## Figures and Tables

**Figure 1 ijms-24-10111-f001:**
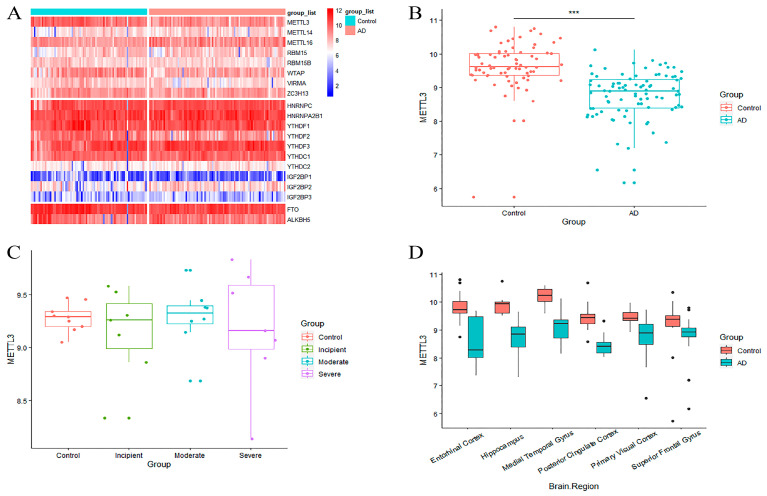
Identification of differential genes between AD patients and controls. (**A**) Heat map of differential expression for 20 m6A regulators in GSE5281, divided from top to bottom according to m6A writer, erasure, and reader genes. (**B**,**C**) Differential expression boxplot of METTL3 in GSE5281 and GSE1297. There were noticeable alterations in METTL3 during the severe stage of AD in (**C**). Patients are categorized as control, incipient, moderate, and severe according to their degree of pathology sequentially for GSE1297. (**D**) Differential expression boxplot of METTL3 in different brain regions in GSE5281. METTL3 expression was markedly decreased in the entorhinal cortex, hippocampus, and medial temporal gyrus. (*** refers to the significance mark of the difference).

**Figure 2 ijms-24-10111-f002:**
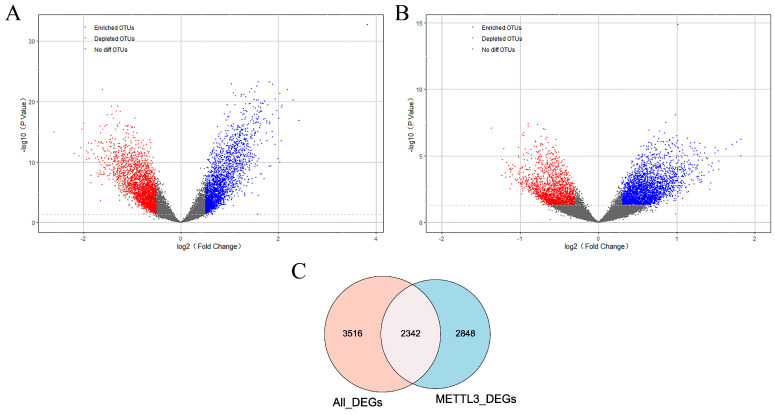
Identification of METTL3-related AD genes. (**A**,**B**) Volcano diagram of DEGs (|log_2_FC| > 0.3 and *p* value < 0.05). DEGs between AD patients and controls and between METTL3_high and METTL3_low subgroups were included. Up-regulated genes were in red and downregulated genes were in blue. (**C**) Venn diagram showed that there were 2342 overlapping genes in All_DEGs and METTL3_DEGs. Namely, METTL3-related AD genes that were correlated with METTL3 and significantly differentially expressed in AD.

**Figure 3 ijms-24-10111-f003:**
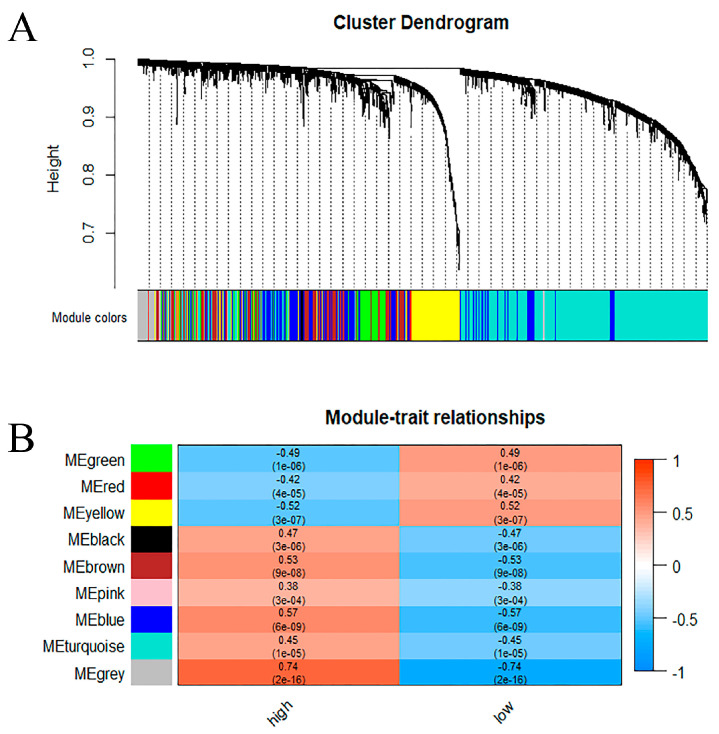
The set of AD genes most positively correlated with METTL3 in AD. (**A**) A dendrogram of METTL3-related AD genes, clustered under different degrees of similarity. (**B**) Heat map of correlation between 9 modules and METTL3 expression status. The numbers were the correlation coefficients and their corresponding confidence levels.

**Figure 4 ijms-24-10111-f004:**
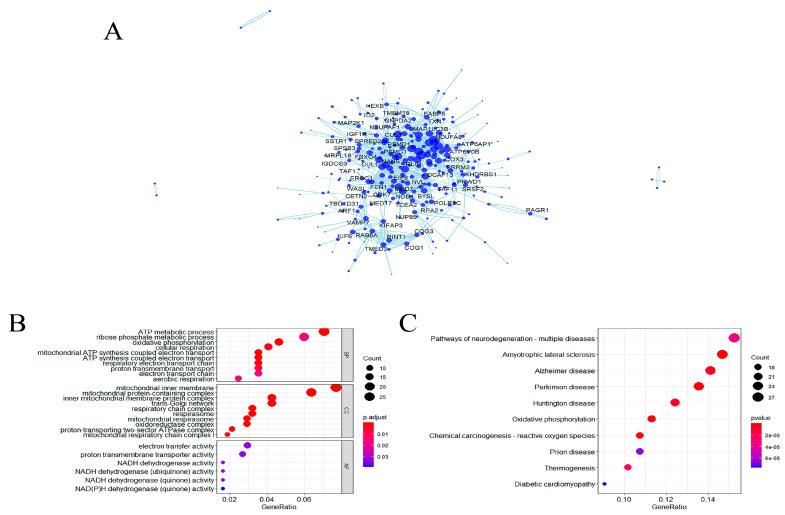
Screening and pathways analysis for the genes that positively related to METTL3. (**A**) The PPI network was constructed from the blue modular genes. Proteins with large circles indicate more linkages around them, where unlinked ones were hidden. The looped circuit among the proteins suggested that they may interact in multiple biological pathways. (**B**,**C**) GO and KEGG pathways enrichment results of the blue modular genes (*p* value < 0.05). Mainly enriched in energy metabolic processes and neurodegenerative diseases.

**Figure 5 ijms-24-10111-f005:**
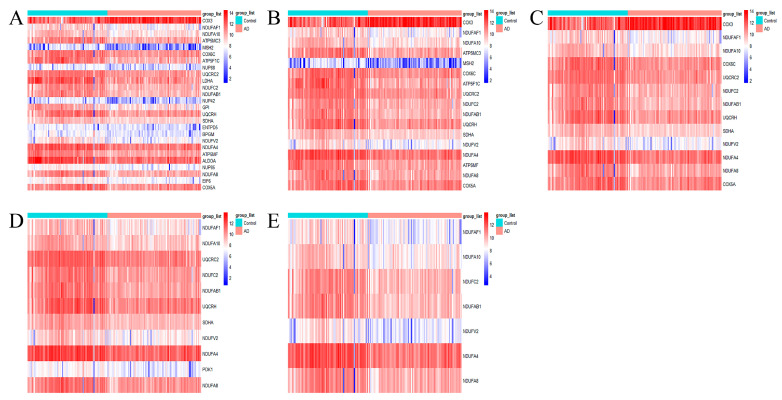
Heat map expression of pathway genes. (**A**–**E**) From left to right, the selected pathways were the ATP metabolic process, oxidative phosphorylation, mitochondrial ATP synthesis coupled electron transport, oxidoreductase complex, and NADH dehydrogenase complex.

**Figure 6 ijms-24-10111-f006:**
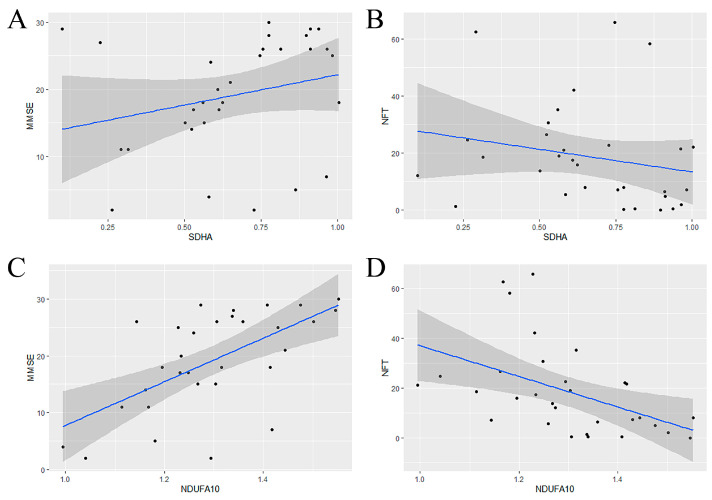
Validating the role of SDHA and NDUFA10 in AD. (**A**,**B**) From left to right, the association of SDHA expression with MMSE and NFT, respectively. (**C**,**D**) From left to right, the association of NDUFA10 expression with MMSE and NFT, respectively. When NDUFA10 was lowly expressed, MMSE values decreased and NFT values increased, indicating aggravation of AD.

**Figure 7 ijms-24-10111-f007:**
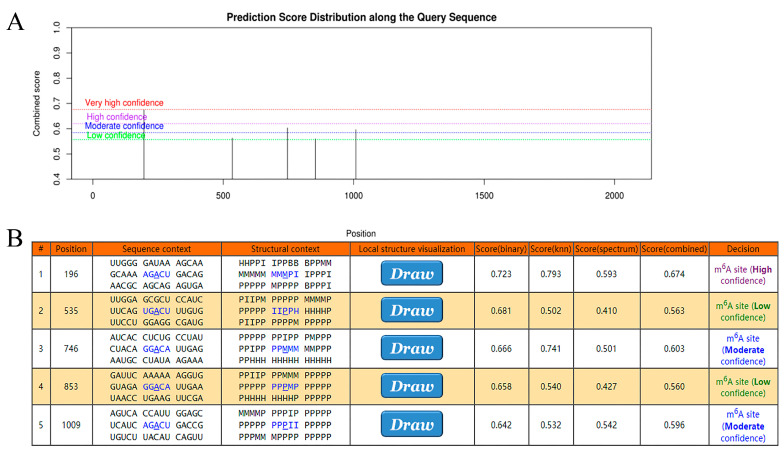
Exploring the relationship between NDUFA10 and m6A modification. (**A**) Distribution of predicted scores of m6A modification sites on NDUFA10. (**B**) Possible positions of modification sites and specific scoring. The blue font and underlined letters represent the base site most likely to interact with METTL3.

**Figure 8 ijms-24-10111-f008:**
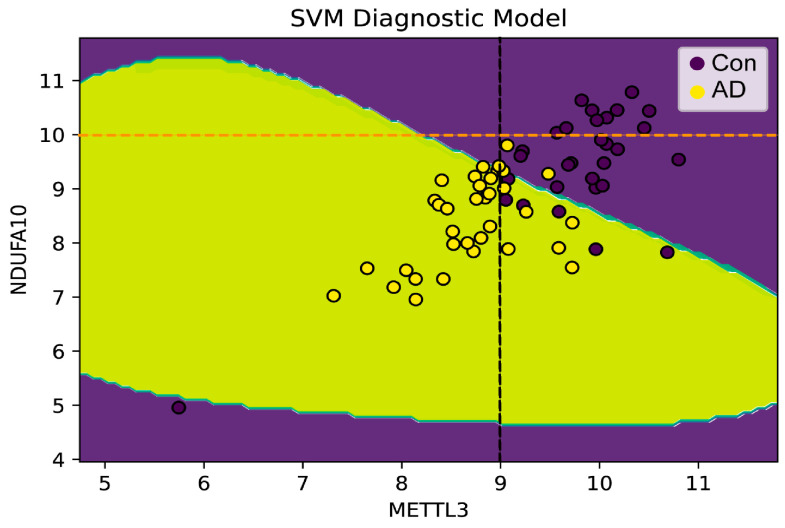
Constructing a diagnostic model of METTL3 and NDUFA10 in AD. The horizontal and vertical coordinates were the expression of METTL3 and NDUFA10, respectively. AD patients were colored in yellow and controls were colored in purple. A threshold for METTL3 expression level was indicated by the black dashed line (x = 9). The orange dashed line indicated the threshold of NDUFA10. The probability of developing AD was higher when both expression levels were below the threshold. Especially, if METTL3 was below the threshold, the probability of risk to AD would be close to 100%.

**Figure 9 ijms-24-10111-f009:**
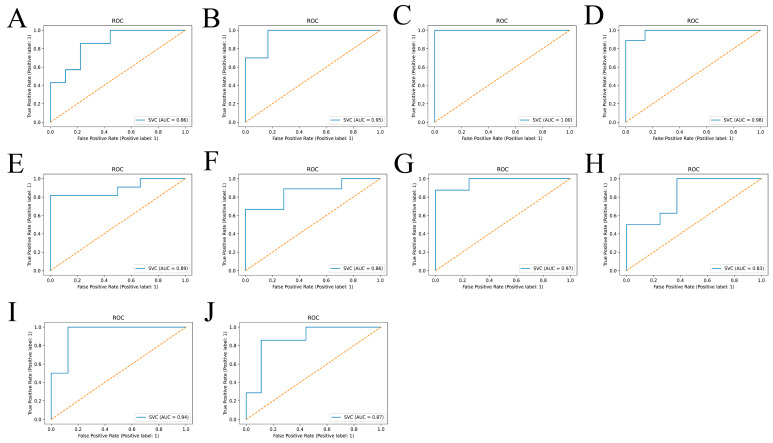
Validation of the model classification effect. (**A**–**J**) ROC curves. AUC represents the area under the ROC curve. When 0.5 < AUC < 1, the model shows a great classification effect and holds predictive value. The training and test sets were randomly divided ten times by cross-validation. The predictive performance of the SVM diagnostic model was evaluated in AD by calculating the average of ten AUC values.

**Table 1 ijms-24-10111-t001:** List of occurrences in the vicinity of METTL3 in 10th gene sequences. Genes with less than 5 were hidden.

Gene	Frequency
SDHA	10
NDUFA10	10
GPI	8
YTHDC1	6
HNRNPC	6
YTHDF2	5
NDUFC2	5

**Table 2 ijms-24-10111-t002:** List of 20 m6A regulators.

m6A Member	Gene Symbol	Mechanisms	References
Writers	METTL3	Enables S-adenosyl-L-methionine binding	[50]
	METTL14	Contributes to mRNA-methyltransferase activity	[51]
	METTL16	Enables METTL3/14 activity	[52,53]
	RBM15/15B	Recruits m6A complex to target sites of mRNA precursor	[54]
	WTAP	Regulates members and handles m6A complex formation	[55]
	VIRMA	Recruits METTL3/METTL14/WTAP to guide methylations	[56]
	ZC3H13	Bridges RBM15/15B and WTAP	[57]
Erasers	FTO	Removes m6A modification	[58]
	ALKBH5	Removes m6A modification	[59]
Readers	HNRNPC	Mediates mRNA splicing	[60]
	HNRNPA2B1	Promotes primary miRNA processing	[61]
	YTHDF1	Promotes mRNA translation	[62]
	YTHDF2	Regulates mRNA stability	[63]
	YTHDF3	Regulates mRNA translation and splicing	[64]
	YTHDC1	Regulates mRNA Splicing	[65]
	YTHDC2	Enhances mRNA translation and decreases mRNA abundance	[66]
	IGF2BP1/2/3	Enhances mRNA stability	[67]

## Data Availability

Gene expression data from the datasets GSE5281 and GSE1297 were downloaded from the Gene Expression Omnibus (https://www.ncbi.nlm.nih.gov/geo/, accessed on 8 March 2022).

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
