# Peer review of "An Exploration of the Coherent Effects between METTL3 and NDUFA10 on Alzheimer’s Disease"

_ijms, 2023, doi:10.3390/ijms241210111_

Round 1

Reviewer 1 Report

The manuscript is well written but incomplite.

1) The introduction section is too short and the information is incomplete.

2) The quality of the figures is poor, it is very difficult to read the figures.

3) There could also be some experiment that would confirm this model.

Reviewer 2 Report

The manuscript entitled "An Exploration of the Coherent Effects Between METTL3 and 2 NDUFA10 on Alzheimer's Disease" is an attempt to explain the pathomechanism of AD development

My comments:

1.       Introduction –

Complement the role of METTL3-related target gene NDUFA10 in the body and their potential mechanism in neurodegenerative diseases.

Describe the relationships between the biochemical parameters studied in more detail.

Too extensively described the purpose of the research.

2.       Results –

Careful and extensive documentation of results.

3.       Materials –

Were other characteristics taken into account when evaluating patients, e.g. age of onset, disease progression, etc.?

Why were these genes selected, and what was the basis for the selection?

How the sections for the study were selected.

4.       Discussion –

“We identified the overall expression pattern of m6A regulators by differential expression analysis between AD brain and normal tissue.” – what is normal tissue in this experiment?

“Specifically, we observed a significant downward trend in METTL3 in AD brain tissue” - In what structures and why?

5.       Conclusion –

“Our findings provide novel insights into how AD can be prevented and intervened with precision” - not very precise. How. To correct.
